# The Curious Case of Bhasan Char: Island Relocation and the Politics of Refugee Containment in the 'Global South': The Case of Bangladesh

Tazreena Sajjad, PhD
Department of Peace, Human Rights, and Cultural Relations (PHRCR)
School of International Service, American University

## 1. Introduction

On October 9, 2021, the Bangladesh Ministry of Disaster Management and Relief, and the U.N. High Commissioner for Refugees (UNHCR) signed a Memorandum of Understanding (MoU) to establish a common framework for humanitarian services for Rohingya refugees on Bhasan Char, an island 37 miles from the mainland in the Bay of Bengal (IISS Myanmar Conflict Map, 2021). The MoU signaled the international community's formal acquiescence to the then Government of Bangladesh (GoB)'s plan to ultimately relocate 100,000 Rohingya from the Kutapalong-Balukhali 'mega camp,' to address what it had described as an 'untenable' situation. It also ushered in a new phase of the narrowing space of hospitality for the Rohingya in Bangladesh, who have sought refuge in the country in successive cycles, with the largest numbers arriving in 2017.

Nested in the literature on the politics of containment and migration diplomacy, this research asks: What factors explain the then GoB's new phase of attempted refugee containment in Bhasan Char? Specifically, given the infrastructural development and investments made in Bhasan Char, which has been hailed as a 'model' for hosting refugees, what explains the choice of an *island* in particular? Finally, what are the broader implications of island relocation with regard to refugee management in the Global South? This study offers three specific explanations relocating some Rohingya to Bhasan Char: (i) repeated failures in Rohingya repatriation and persistent international indifference to the protracted crisis; (ii) perception of the Rohingya not only as sources of political and economic insecurity, but as a threat to the local environment and land; and (iii) the history and political economy of  khas (public) land use  - including islands - in Bangladesh.

This research aims to make two main contributions. First, in identifying the Rohingya as a perceived threat to land access, and the context of the island relocation process,, it draws attention to the complex sociopolitical role of land in Bangladesh. The literature on refugee management in the Global South is expanding (see e.g. Hollifield and Foley 2022; Natter and Thiollet 2022), together with research examining the ability of such states to instrumentalize forced migration for their domestic and foreign policy goals (Micinski 2021; Tsourapas, 2019; Tennis 2020). There is now literature on the use of camps and borders as a means of immobilization, which have also produced encounters, resistance, or conditions for competing sovereignties (Kodeih, et. al 2023; De La Chaux, et. al 2018; Dalal, 2020). However, there is limited literature on *why* specific campsites may be selected outside of the fact that they allow for the separation of the 'citizen' from the 'outsider.' Such explanations however do not fully explain the ongoing presence of a greater number of Rohingya in the Cox's Bazaar camps, and the selection of Bhasan Char, where the GoB made substantive investments to make the island more hospitable. Neither do they capture how land is used in Global South contexts such as riverine Bangladesh, where island residence is an integral part of its historical, political and social landscape. In drawing attention to Bangladesh's land use and its politics, this project emphasizes the need to understand the use of an island for refugee management from a Global South perspective, as opposed to Global North's off-shoring practices.

Second, this research draws attention to how the Global South – arguably producing the highest number of number of labor migrants and with long histories of hosting forcibly displaced populations - strategize and exercise agency in refugee management. Scholarship on migration diplomacy – how states bargain and negotiate their interests vis-à-vis migration within diverse contexts in Global South - is now burgeoning (Norman, 2020; Malit and Tsourpas, 2021; Adamson and Greenhill, 2023). Fernández-Molina and Tsourpas (forthcoming) furthermore offer the framework and typology of 'migration power' to capture ways in which it is exercised, and how it

manifests across asymmetrical North-South lines and state-non-state actors. Such a framework is helpful in reflecting on how, for instance, Bangladesh - a country in the Global South - with limited bargaining power, can combine mechanisms of containment and discourses of securitization to negotiate with the donor community. At the same time, such a framework does not engage with context-based 'solutions' that individual countries may implement in an attempt to strengthen their negotiating position.

The power assymetries between Global South contexts and the geopolitical value of forcibly displaced people are significant factors in negotiations around migrant management. Tsourapas (2019) introduces the term refugee rentier states to explain how 'important' large hosts such as Turkey extract rent from powerful state actors to maintain refugees within their borders. His refugee rentier model, however, does not unpack the geopolitical importance attached to specific categories of refugees and certain hosts, and is limited in its applicability to countries that share borders with more powerful counterparts. Micinski (2021) refines the model to explain how contexts like Kenya and Pakistan that do not have geographical proximity to powerful countries use the threat of expulsion to to extract refugee rents. However, the framework does not explain why poorer states have not leveraged their ability to expel refugees to gain more access to rents, even when they struggle with refugee-hosting 'exhaustion.' Frier et al's framework of refugee commodification (2021) is helpful in examining how states – irrespective of geographic location and economic positionality - learn to instrumentalize refugees for gains. However, it is also limited in explaining contexts where possibilities of revenue generation using refugees are limited, and the localized dynamics of where containment may happen.

The selection of Bangladesh as a case study – where refugee relocation is taking place *within its own borders* – is therefore important at multiple levels. First, while Bangladesh shares a border with a regional and nuclear superpower – India – it is *not* using the Rohingya as a leverage to extract rents from its neighbor. Second, unlike Kenya or Pakistan, it is not threatening mass expulsions to obtain more international aid.. Third, compared to Ukrainians and (to some extent) Syrians and even Afghans, the Rohingya remain a refugee community of 'low' geopolitical value, who garner limited international interest – a reality that has continued to hamper Bangladesh's efforts to generate support to facilitate their sustainable repatriation. It is within these constraints that the GoB's calculations for Bhasan Char - drawing on its historic practice of using public landfor refugee management– needs to be examined. However, the Bhasan Char strategy, while garnering international attention, has not produced substantive material benefits. Instead, Bangladesh continues to absorb significant portion of the financial costs to maintain a remote refugee camp, and questions of return and belonging for the Rohingya remain unresolved.

## 2. Methodological Approach

Using a single case study of Bangladesh, this article draws on original materials and interviews from multi-sited fieldwork conducted in Dhaka and Cox's Bazaar in 2022 with academics, political elites, government officials, and international and national NGOs working with the Rohingya population follow-up interviews between in 2023 and 2024. To complement the interviews, I conducted a textual analysis of reports, public statements and documents produced by international and national NGOs, the UNHCR the GoB, former PM Sheikh Hasina's international and UNGA speeches, and Rohingya coverage in prominent Bangla and English newspapers between 2017-2023. The textual analysis centered on searching for specific references to the Rohingya and Bhasan Char, and identifying key words and tropes that frequently occurred.

The single case study allows for 'an empirical enquiry that investigates a contemporary phenomenon in depth and within its real-life context, especially when the boundaries between phenomenon and context are not clearly evident' (Yin, 2009: 14). It also provides rich detail and a *situated* analysis that aims to represent the complexity of the object of interest. As such, the single case study approach constitutes a reservoir of local knowledge within a 'specific, unique and bounded system' (Stake, 2008: 443, 445), useful in understanding the social world thorough analysis of the particularistic nature of the distinct phenomenon. The focus allows for a deeper understanding of the sociopolitical dimensions of hosting *stateless* refugees from a protracted situation in a 'developing' country. Moreover, it allows for a thorough examination of a core finding of the project – Bangladesh's political economy of land use, and in particular islands - within which the relocation program has to be contextualized. Such an analysis reveals an understanding of relocation not as a form of punishment, as has been the case where refugees have been contained in islands (e.g. Moria in Greece) or responsibility transference (e.g. Australia's use of Nauru and Papua New Guinea's Manus Island), but as a means of addressing specific concerns of land shortage and environmental constraints within the Bangladesh context.

The paper is structured as follows. First, I situate the study in the broader literature, followed by a brief historical background of the Rohingya in Bangladesh. Second, I discuss each of the three findings that explain the GoB's decision to relocate a small percentage of the Rohingya to Bhasan Char, focusing particularly on the political economy of land use in Bangladesh. I then delves into how island relocation considered a 'winning' calculation by the former GoB, has in fact generated significant criticism and limited financial dividends for Bangladesh. As such, I argue that the case of Bangladesh exposes the limits of Tsourapas' refugee rentier state, where a host country can employ different strategies to leverage its demands from the international community. Rather, the outcome – greater isolation of the Rohingya and financial burden of maintaining the island – presents a failure of GoB's strategic calculation. In conclusion, the research raises questions about how remote relocations by Global South hosts can produce an outcome that leaves issues of belonging and citizenship unresolved for many of the world's displaced, while benefiting the ongoing containment policies of the Global North.

## 3. Situating the Research
Once the Cold War ended, the refugee no longer possessed ideological or geopolitical value for the 'west.' By the 1990s, the foundations for a paradigm shift in international refugee policy and law was laid *inter alia* through the creation of the 'myth of difference': whereby the nature and character of refugee flows in Europe constructed as 'white, male and anti-communist,' was held up as a striking contrast to the 'Black and brown' masses fleeing the 'Third World' (Chimni, 1998). The seamless framing of these latter conflicts as intractable, primordial, and disconnected from the legacies of colonization, successive military or covert interventions, and the violent exploitation of the global political economy further cemented the otherization of the irregular migrant as lacking 'European/western' values, while reducing the space for asylum in the Global North. This waning of interest meant that dependence on poorer countries to contain the forcibly displaced increased multifold. In this dichotomy, responsibility-sharing became framed as a choice for the former, shaped by highly selective refugee resettlement programs and a complex asylum system for the very few, and the de facto *duty* of the latter. As such, the North, based on its own geostrategic calculations, may or may not take the incentive to share the responsibility, producing what Betts (2009) called the 'North-South impasse.' In pragmatic terms, this translated into the imperative of poorer countries having to

absorb and negotiate the politics of hosting large populations. Meanwhile their wealthier counterparts as UNHCR donors may offer *or* cut international assistance based on their interests and constraints, while remaining shielded from the 'disorderliness' of refugee life. The logic then follows that the Global South 'has little choice other than to host refugees' (Betts, 2009: 13). Consequently a powerful myth has gained broad salience -that refugee crises are regionalized and a unique problem *of* and *by* the Global South, not a reflection of the failures of the international system, or a product of the state-system itself, nor produced in many instances by military, economic, and political interventions of the Global North.

These two dynamics – reduction of the space for refuge and increasing reliance on the Global South for 'warehousing' have produced two distinct realities. First, in the Global North, undocumented migrants increasingly face raids, detention, and deportation when assessed to fall below the extreme standards set for being considered deserving of protection (see e.g.Korvensyrjä 2023; Moncrieffe and Eyeben, 2007; Sajjad, 2018; Hamlin, 2021). Second, by manipulating ongoing power inequalities, the Northern penal state (Aas, 2013) is adopting measures that various scholars have theorized as 'border externalization,' 'remote border control' and 'extra-territorialization' (Hyndman and Mountz, 2008; Welander, 2020). Such strategies have included exporting penal models and technologies of crime regulation, outsourcing border control apparatus through financial and military incentivization to migrant 'sending' or 'transit' countries, and offering development assistance in return for migration containment – in short, transforming the latter into the North's migrant gatekeepers (Frowd, 2020; Lee, 2022; Chemlali, 2023). These efforts further consolidate the long established practice of warehousing refugees in and around their places of origin – through the use of the refugee camp.

*Where is the Global South?*
If walls, drowning bodies, biometrics, and detention centers are the visual markers of irregular migration at the northern borderlands, our collective geopolitical imagination conjures up the refugee camp as the main entry point for engaging with refugees in the Global South. In this imagery, tents, nameless brown and Black bodies, and desolate terrain define the 'overcrowded' landscape of refugee-hosting. This is contrary to the reality of variation in camps in places as diverse as Uganda, Pakistan, Jordan, Turkey, Kenya, Lebanon, and Palestine. Such a geopolitical visualization also decries the camp's historical trajectory, which began in the 'west' (Forth, 2015; Bartov, 2003).Today, the institutionalization of the refugee camp has been naturalized as a 'product of failed international idealism and political expediency and serves as a timeless reminder of the plight of displaced peoples' (Tusan, 2021). Frequently characterized with demarcations (fences, barbed wire), it remains the visual marker of spatial containment and a core components of refugee governance - despite the growing reality that the majority of the world's refugees live in urban areas.

While extensive scholarship now exists about the architecture and life in refugee campsas spaces of extraterritorality, carcerality and exclusion despite various levels of porosity (Agamben, 1998; Ramadan 2013; Agier 2011; Turner and Whyte, 2022; Turner, 2016), there is limited research on the choice of selecting the *location* of a refugee camp. The existing literature on population transfers in history in an effort to create ethnically specific settler spaces to make ethnicity, religion, and race central to both citizens' and subjects' relations with the state is instructive -although not fully explanatory - in this regard. 19[th] century European colonial practices strategically pushed particular indigenous populations out of colonized territories, sometimes with genocidal violence, in the interests of creating racially specific landscapes (Moses, 2010).  Some examples include the French granting special citizenship rights to Algerian Jews and settling Europeans on land appropriated from

Muslim Algerians; the British dispossessing indigenous inhabitants in parts of Africa and Australia to create white settler colonial enclaves; Zionism's role in resettling a minoritized population in Palestine; and the 'repatriation' of African-Americans to Africa (Robson, 2020). Chatty (2010)'s work focuses attention on the involuntary resettlement of diverse communities in the Middle East, situating their experiences within the Ottoman Empire's internal power struggles, border wars with its neighbors, western colonization, and 20th-century Arab nationalism, leading to the formation of cosmopolitan identities (Ibid, p. 6). Hamed-Troyansky (2024) takes this understanding further, arguing how the Ottoman government – pre-dating the League of Nations - developed a refugee regime through which the latter resettled North Caucasian refugees throughout the empire, while economically supporting these newly established villages and consolidating state authority.

Contemporary critical scholarship on refugee containment in the Global North has drawn attention to the consolidation of the 'crisis' narrative, the strategic weaponization of natural environments (land, sea, deserts), the criminalization of solidarity work, increased use of militarized border technology (satellite surveillance systems, biometrics, robotic dogs) for migrant deterrence and their violent consequences (see e.g. De Leon, 2015; Slack and Martínez 2020; Solano and Massey, 2022; Ackelson, 2005). In Europe, the Mediterranean has become a 'carceral landscape' (Stierl, 2021) where pushbacks and maritime deaths are routine (Dickson, 2020; Kinacioglu 2023). In addition to mountains, forests in Europe too have been landscapes of deterrence – either through deforestation to deter entry, or as a means to build technology for ground and aid surveillance (Hameršak and Pleše, 2020). Australia's practice of using Nauru and Manus for offshoring detention practice has also produced considerable literature (see e.g. Barnes, 2022). In *The Death of Asylum*, asking 'What is a prison if not an island?' Mountz (2020: 20) focuses on how the US, Canada, Australia, and the EU strategically use the island *as the border*, isolating people trying to seeking asylum. However, the literature on the history of strategic ethnic compartmentalization and population resettlement does not fully capture the rationale for contemporar refugee containment in the Global South, particularly since they do not seamlessly follow the colonial logic of conquest and separation. Similarly, the scholarship on the weaponization of natural terrain for migrant deterrence in the Global North, including the use of islands, does not capture the historical, social, political dynamics that may inform a Global South context's decision to use specific locations for refugee containment. It is here that this research aims to make a contribution.

### 4. Failed Repatriation Efforts, International Indifference, and Rising Resentment: Creating the Logic for Bhasan Char

The *Rohingya* are a highly marginalized ethnic minority – largely, albeit not only, Muslim – in the northern Rakhine State of Myanmar. Rich in oil and natural gas reserves, and the site for vested political and economic interests for Myanmar, China, and India, nevertheless it has remained the country's least developed state with an 80 percent poverty rate even prior to the COVID pandemic (UNDP, n.d.). The deeply contested question of the origins of the Rohingya in which British colonial rule played a critical role, and their unfulfilled promise to create a separate 'Muslim National Area' in return for Rohingya support in World War II, have been critical in setting the stage for ongoing ethnic tensions in Myanmar (Uddin 2020). These tensions have resulted in periods of state-sanctioned violence and discriminatory laws against the Rohingya; since the passage of the1982 Citizenship Act of Myanmar, they have also been denied their right to citizenship (see e.g. Brett and Hlaing, 2020). This last policy is in line with what every Government of Myanmar (GoM) has insisted –- that the land had always been exclusively Buddhist, the Rohingya as an ethnic group never existed, and that those who identify as such are 'illegal *Bengali* migrants' (Ibrahim, 2016; Ullah and Chattoraj, 2018). Correspondingly, the Rohingya have remained a convenient scapegoat for the political ambitions of Myanmar's military generals and politicians, and a diversionary distraction from domestic crises subject to forcible displacement since 1942; mass expulsions in 1962, 1970, 1991; and most recently in 2017 and beyond (Stokke, et. al, 2018).

Over the years, Rohingya fleeing ongoing persecution arrived across south and south-east Asia and Saudi Arabia. The largest host since 2017 with over 1 million Rohingya, Bangladesh received significant numbers of Rohingya in two previous cycles. The first was in 1977, following Myanmar's *Operation Nagamin* (Dragon King), a brutal military and immigration operation to register citizens and screen out foreigners (Elahi, 1987). The second cycle was between 1991 and 1992 during *Operation Pyi Thaya* (Clean and Beautiful Nation) when over 250,000 Rohingyas fled Myanmar (Kaveri and Rajan, 2023).

Rohingya reception in Bangladesh has been complex, produced at one level by co-ethnic solidarity[i] (although not all Rohingya, nor are all Bangladeshis Muslims; and neither are *all* Bangladeshis Bengalis – the closest ethnic overlap with the Rohingya),[ii] fluid border identities, family ties, and historically limited border controls. At different times, Rohingya rejection has been shaped by funding limitations and cross-border security challenges, which have scapegoated refugees particularly when armed insurgents have operated within camps. As a non-signatory to the 1951 Refugee Convention, Bangladesh has consistently insisted that an early, voluntary, and sustainable repatriation is the country's priority when accepting each large wave of Rohingya arrivals. In fact, in all official statements, interviews and in the PM's UNGA speeches between 2017-2023, there has been an emphasis on the Rohingya being *Myanmar nationals*, that Myanmar is their *homeland*, and their stay in Bangladesh is temporary (Sheikh Hasina's UNGA speeches 2017-2023; Voice of America, 2017). Furthermore, the classification of the Rohingya as 'Forcibly Displaced Myanmar Nationals,' was strategic given that it counters the insistence by successive Myanmar governments that as illegal *Bengali* migrants, the Rohingya have no claims upon Myanmar (Uddin, 2020).

Since 1978, Bangladesh has been party to several separate bilateral frameworks with Myanmar to facilitate Rohingya repatriation. The 1978 Bilateral Agreement on Repatriation of Refugees from Bangladesh to Myanmar facilitated the then newly formed GoB, already struggling with the aftermath

of the 1974 famine and the return of 10 million refugees from Bangladesh's brutal war of independence in 1971 to repatriate 187,000 of the more than 200,000 Rohingya arrivals (Abrar, n.d.; Reid, 1994). The 1992 Joint Understanding with the State Law and Order Restoration Council (SLORC) paved the way for a series of large-scale coercive repatriation programs between 1993 and 1997 until UNHCR brought them to a close (Human Rights Watch, 1993). By 1998, the qualifications for being returned were also made stricter based on whether the then SLORC could verify potential returnees' residence; consequently, repatriation became often impossible.

In light of the most recent arrivals, on September 11, 2017, the Bangladesh Parliament passed a unanimous resolution denouncing Myanmar for atrocities. In submitting the resolution, the then chairman of the Parliamentary Standing Committee on Foreign Affairs Ministry stated

> the UN and the world community should be urged to exert strong diplomatic pressures on Myanmar government for stopping continuous repression on… the Rohingya…, …[and] ensuring their safe accommodation…and giving them rights of citizenship (The Daily Star, 2017).

In an effort to initiate a repatriation program for the newest arrivals, Bangladesh and Myanmar finalized the 2018 Arrangement on the Return of Displaced Persons from Rakhine State drawing on the 1992/93 Agreement, based on which the first round of returnees including 2,260 Rohingya were scheduled for return (Human Rights Watch, 2018). However, fear of ongoing violence together with confusion about who were cleared for repatriation did not lead to a successful return process. In addition, Bangladesh also signed the 2018 Memorandum of Understanding (MoU) with UNHCR relating to the voluntary returns of Rohingya refugees. In August 2019, another attempt was made towards repatriation, with a list of 55,000 Rohingya prepared and initially 3,450 cleared for repatriation (UNHCR, 2019). However, the Rohingya refused to comply, demanding accountability for those responsible for the 2017 atrocities, assurance of full citizenship rights, and return of their seized assets from the GoM. Between 2017 and 2019, while engaging in talks with China, India and ASEAN, the GoB presented three separate plans to the UNGA demanding an end to the ethnic cleansing in the Rakhine state, the need to address the root causes of the crisis, and requesting international assistance for sustainable repatriation. Furthermore, while it was excluded from the 2018 Tripartite MOU between UNHCR, UNDP and the GoM, Bangladesh joined Myanmar and China on the Joint Working Group for Rohingya repatriation. It also signed the the 2021 MoU with UNHCR that focused exclusively on Bhasan Char and worked with the GoM on the China-brokered the 2023 Pilot Repatriation Plan.

**FIG I: KEY AGREEMENTS ON ROHINGYA REPATRIATION**[iii]

| Year | Bilateral Agreement | Parties |
|---|---|---|
| July, 1978 | Bilateral Agreement on Repatriation of Refugees from Bangladesh to Myanmar | • People's Republic of Bangladesh<br><br>• Socialist Republic of the Union of Burma |
| April, 1992 | Joint Understanding on Repatriation of Refugees from Bangladesh to Myanmar | • People's Republic of Bangladesh<br><br>• State Law and Order Restoration Council (SLORC) of Burma |
| November 2017 | Arrangement on the Return of Displaced Persons from Rakhine State [Based on 1992/93 Agreement] | • People's Republic of Bangladesh<br><br>• Republic of the Union of Myanmar |
| April 2018 | MOU Relating to Voluntary Return of Rohingya Refugees; | • Government of Bangladesh<br>• UNHCR |
| May 2018 | Tripartite MOU | • UNHCR<br><br>• UNDP<br><br>• Government of the Republic of the Union of Myanmar |
| March 2023 (ongoing) | 2023 Pilot Repatriation Plan (mediated by China)) | • Government of Bangladesh<br><br>• Government of the Republic of the Union of Myanmar |

In mid-2023, a delegation from GoM conducted interviews with Rohingya in the camps to authenticate their claims (Strangio, 2023). A total of 1,140 Rohingya were slated for repatriation, and a delegation of 20 Rohingya, accompanied by seven Bangladesh government officials, visited two of Rakhine State's 15 villages as a 'confidence-building measure' for voluntary repatriation. Such efforts came under criticism given that the conditions on the ground in Myanmar have not changed.. Tun Khin, President of the Burmese Rohingya Organisation accused the international community of playing 'ping pong' with the Rohingya, saying that this latest attempt 'is a public relations process [by] governments while core issues of the treatment of Rohingya by the Myanmar military are ignored' (Paul, 2023).

Since 2019, not a single Rohingya has been repatriated. While Bangladesh has largely respected the international principle of non-refoulement, there have been allegations of coercion in the Rohingya meeting with Myanmar junta officials and concerns that an inaccuratepicture of the the 'transit camps' had been produced to encourage repatriation. with In addition, security forces were reported to have increased surveillance of Rohingya on the pilot repatriation list (Human Rights Watch, 2023). Meanwhile, Bangladesh has remained frustrated with Myanmar. According to Bangladesh's Refugee Relief and Repatriation Commissioner, 'The process of verification [has been] very slow…the Myanmar authorities oscillates between granting [the Rohingya] their demand to return and refusing it… they say one thing and then another' (interview, 2022).

The combination of the 2021 crisis in Afghanistan precipitated by the US withdrawal, Covid-19, and western attention being diverted to Ukraine following the Russian invasion, have also meant international attention on the Rohingya has significantly diminished. Inter-Sector Coordination Group (ICSG) representative Arjun Jain noted, 'given that it has been over five years, and both the GoB and civil society's response has demonstrated capacity to manage the camps, the Rohingya situation in Bangladesh is no longer a crisis' (interview, 2022). The combination of these factors may explain why international assistance for the Rohingya in Bangladesh has been dwindling (Sajjad, 2022a). For instance, only a quarter of the 2023 Rohingya humanitarian crisis response plan, that outlined a need of $875 million was funded (United Nations Office for the Coordination of Humanitarian Affairs, 2023). The World Food Program (WFP) also significantly slashed its Rohingya budget; in 2023, the value of the food vouchers for camp residents was reduced from $12 per person per month to $10, and in June, to just $8; the equivalent of 27 cents a day (United News Bangladesh, 2023). [iv] Meanwhile, frustrations in Bangladesh including among the local community have grown. In response to a 2020 UN statement on Rohingya relocation as a means to decongest the camps a seemingly annoyed spokesperson for the Foreign Ministry asked: 'Where were the UN authorities when the Rohingya people were forced to leave their ancestral homes in Rakhine of Myanmar amid atrocities?' (New Age, 2020). Similarly, a civil society actor interviewed in Dhaka remarked:

> it's the decision to open our doors has ushered in a danger – thanks to Myanmar's position and international indifference – that now stalks Bangladesh. The Rohingya have become our problem now.

Such a position was also reiterated by higher political officials. In 2020, the Bangladesh FM hailed the decision for island relocation to be a prudent one stating:

> While the global leadership and the UN agencies have been extending lip service to the persecuted people of Myanmar…, none came forward either for their relocation or sending them back to …Myanmar. Furthermore, investment trade …from Europe, ASEAN countries, China, Japan and the UK have only increased …. none of the human rights organizations put any blockade to those countries…They did not even ask them for divestment as they did in the case of Apartheid in South Africa (Bangladesh Post, 2020).

## 5. The Dimension of Security: The Rohingya as Multiple Sources of Threat

The Rohingya in Bangladesh have increasingly been framed as a traditional source of insecurity related to terrorism, narcotics, and political instability. Rana and Riaz (2022) argue that Rohingya securitization in Bangladesh commenced as early as 1992 following the failure of coercive repatriation efforts with the framing of Rohingyas as 'aliens,' and 'illegal economic migrants,' emerging in both speech acts and non-discursive strategies. Such efforts intensified in subsequent years, notably amidst

the sectarian violence that characterized the post-2012 landscape in the Rakhine state during which Bangladeshi authorities implemented a push-back policy, sealing both land and sea borders. The 2013 adoption of the National Strategy on Myanmar Refugees and Undocumented Myanmar Nationals played a critical role in formalizing the securitization process.

Despite the 2017 'open-border' policy and the framing of the new arrivals as reminders of mass displacement in 1971, (Sajjad, 2022 b), militant Rohingya leaders' long-standing relationship with Bangladesh's largest Islamist political party – the Jamaat-e-Islami, (JI), whose activities remain a source of deep contention in the country - have continued to make the Rohingya camps a site of intense scrutiny. Since 2019, there has been an intensified focus on insurgent recruitment, narcotics trafficking, arson, kidnappings, targeted killings of activists and *majhis* (camp leaders), and turf wars in the camps (Karim, 2019; International Institute for Strategic Studies, 2023). Such a focus has contributed to the narrative of the Rohingya *themselves* being a security threat, justifying the deployment of checkpoints in Cox's Bazaar, imposing restrictions on mobility and telecommunications; and the installment of barbed wire fencing around the camps.

The framing of the Rohingya as a threat to national *and* regional security on an international forum arguably first found traction in a joint program organized by the policy groups of Bangladesh and India (UN News Bangladesh, 2019). The former PM's 2022 UN General Assembly statement reflected this shift:

> Prolonged presence of the Rohingyas in Bangladesh has caused serious ramifications on the economy, environment, security, and socio-political stability in Bangladesh…Cross border organized crimes including human and drug trafficking are on the rise. Even, this situation can potentially fuel radicalization. If the problem persists further, it may affect the security and stability of the entire region, and beyond.

Again, at the 2023 UNGA session, Hasina emphasized that the Rohingya situation in Bangladesh had continued to generate insecurity in the country (Bangladesh Sangbad SangsthaNews, 2023). Nakajima, with the Medicins du Monde Japan noted that

> the GoB's increasing pressure on Rohingya for repatriation and strengthening has been accompanied by its narrative of security in the local area, national and the region, especially since the 2019 Genocide Commemoration Day, where over 200,000 Rohingya, demanded justiceafter the second failed repatriation attempt (Interview, 2022).

The narrative of the Rohingya as a threat mirrors ongoing securitization of the irregular migrant/refugee/asylum-seeker and a *societal* problem in the west increasingly requiring control through surveillance and risk management (e.g. Huysmans 2006; Slootweg, et. al 2019; Mountz, 2020). In the last three decades, the securitization of forced migration has particularly become noticeable in the Global South. The need to protect communities and their entitlements against *different* outsiders who are an existential threat to communal identity, rights, and privileges has become more pronounced with the protracted nature of hosting refugees. Yet, there is limited scholarship on how the framing of the refugee as a 'national security threat' is finding traction in the Global South, and galvanizing counterterrorism policies and violence against refugees (see e.g. Brankamp and Glück, 2022). Wilkinson (2007), and Ayoob (1997) have argued that the predominant approaches within critical security studies are intrinsically limited in their engagement with the diversity in Global

South such that while applicable in North American immigration contexts, its effectiveness in non-Western settings remains questionable. Together with other post-colonial scholars, they have argued that securitization theory maintains a European bias and remaining confined to a narrow range of identified threats coupled with assumptions of Western democracy that tend to linearize the securitization process (Wilkinson, 2007). By erroneously assuming full availability of threat-related information and decisions within the public domain, it overlooks how in the Global South broadly, such information may not be publicly accessible; and how decision-making regarding security threats is characterized by less systematic and more ad hoc processes. Nigusie and Cheru (2022) also contend that the pivotal role of Global South host states and their negotiations around multi-level pressures are overlooked in the prevailing literature on securitization. Furthermore, focusing on south-east Asia, Jones (2011) draws attention to its limitations given its focus on discourse rather than practice, and how it does not take into account historical and social structures, and power struggles shaped by colonial legacies and state-led economic development under the international liberal order (Jones, 2011).

The narrative of the Rohingya as a 'threat' therefore demands contextualized *within* the peculiarities of socioeconomic constraints in Bangladesh. In 2017, the former Finance Minister feared that the Rohingya would be a big pressure to the country's economy, while driving away tourists from Cox's Bazaar – a major tourist attraction - and said that 'Myanmar will destroy Bangladesh's economy this year' (Badal , 2017). While such alarmist rhetoric has not been materially substantiated, there have been negative economic impacts in the local context. These included price hikes of food items, increased competition in the local labor force with the Rohingya charging cheaper wages (Ansar and Khaled, 2021; Alam, et. al, 2023); and higher costs of living as a result of the presence of national and international aid workers in the area (Sajjad, 2022b). This is despite the fact that while Rohingya-run enterprises face greater challenges than their local counterparts, and are smaller and less profitable (Filipski, et. al, 2021). In addition, the Rohingya are increasingly seen to be a demographic concern in Cox's Bazaar. In 2020, the Bangladesh Foreign Minister statement in explaining the rationale for Bhasan Char noted,

> There are more than one thousand Rohingya children born in the camps every year which are getting increasingly congested..there is a growing sense of desperation amongst the Rohingya, resulting in a deteriorating law and order situation in the camps. It is because of this…that the GoB has been forced to take on the financial responsibility for, and arrange the relocation to Bhasan Char… (Maksud, 2020).

Azizul Hoque with the Refugee Unit of BRAC University connected the issues of the demographic concern, insecurity within the camps, and economic questions to the ongoing concern about land shortage and ownership, which remain central to the sociocultural and economic identities of Bangladeshis and define their access to power. He noted,

> Once it became clear that the Rohingya were not leaving, there was a fear that there would be a greater Arakan with the Rohingya claiming more land – something already in short supply - and intensifying competition for the existing labor force in Cox's Bazaar where they already outnumber the local population (Interview, 2022).

Last, but not the least, Rohingya presence in Bangladesh have been increasingly seen to be a source of environmental pressure and ecological imbalance. At the 2020 Climate Vulnerable Forum Leaders event, the former PM said,

My country is facing recurrent flooding this monsoon causing immense damage to crops and displacing huge people…The 1.1 million Rohingya refugees from Myanmar given shelter at Cox's Bazar are also causing serious social and environmental damages.

These references to camp congestion and environmental pressures were increasingly peppered in the PM's speeches. While environmental challenges cannot be solely placed on the newest arrivals, research finds that there the large number of arrivals have stressed the local environment. For instance, Sakib's (2023) research and Sarkar et. al's (2023) studies found that the 2017 arrivals created a severe pressure on the local environment in Cox's Bazar when most of the cultivable lands, hills and forestlands were occupied for settlements, as refugees frequently accessed natural resources. Despite efforts underway to increase positive interactions between the environment and surrounding communities, access to ecosystems continue to fuel socio-economic tensions between the locals and the Rohingyas – and have served as an explanation for the need to relocate some Rohingya elsewhere.

## 6. Bhasan Char in the Context of Bangladesh's Political Economy of Land Use

Across the Global South, land remains an imperfect commodity, not yet fully integrated into capitalist social relations of production as a form of inalienable private property yet acting as an asset that substitutes for all costs associated with welfare. This means land – as a productive asset and a cultural practice – remains a material, institutional, and ideological site of intense contestation by multiple stakeholders, including governments and local communities (Lombard & Rakodi, 2016).

In riverine Bangladesh, where over 10 million people live on sedimented islands, char (island) living has to be contextualized within the political economy of land use and interests associated with agricultural production and social security of vulnerable groups. Historically, char living has been shaped by disputes, violent conflicts, local, and regional influence. Furthermore, the broader dynamics of the political economy of land use in Bengal have produced a tense and coercive relationship between governments and local farmers (Haque, 1997). Since the 1793 Permanent Settlement, in which the British determined the local owners of the land and the fixed revenues they could collect, the Bengal charlands were under the legal jurisdiction of the state or certain individuals, with no satisfactory provisions for the re-settlement of people displaced by river erosions and flooding along the chars. Post-British independence, land-based power politics was shaped by asymmetrical 'patron-client' or 'headman-subordinated follower' relations (Zaman, 1996). Such dynamics dominated the lives of marginal and landless farmers in the charlands (Sarker et. al, 2003).

Following Bangladesh's independence, recognizing the challenges in the charlands, the government reclaimed the control of alluvial lands from the *jotedars* (large landlords) and redistributed them among the landless and small landholders (Haque and Jakariya, 2023). The 1972 Presidential Orders was critical in allowing the government to claim chars as khas (public) land (Haque,1997). Some of this land was used to rehabilitate those who were impacted by severe river erosion, laying the groundwork for the Ashrayan (shelter/accommodation) initiative - a social security program for landless populations of tremendous significance in later decades - and which aligns with Bangladesh's commitment to the UN sustainable development goals.

The 1982 Land Acquisition and Requisition of Immovable Property Ordinance - rooted in the 1894 British Colonial Land Acquisition Act used to colonize unsettled lands and collect revenue - and the

1989 Act, which paved the way for the GoB to acquire property, also allowed the GoB to acquire khas land for large-scale infrastructure, development projects, and resettlement schemes in the face of flooding and erosion (Zaman, 1996). To address ongoing concerns about the power of elites in the charlands (Tariquzzaman and Rana, 2014), modifications were made to alluvial land tenure policies such as the 1994 amendment to the 1972 Presidential Order No. 135, which meant - among other issues - that land that re-emerges after three decades becomes government-owned. In 1997, the Agricultural Khas Land and Settlement Policy was also enacted to distribute khas land to the landless on 99-year leases (Masum, 2017). These policies continue to impact those who live in the charlands, who also struggle with annual flooding during the monsoon season, weak health infrastructure, and limited economic opportunities. Consequently, many char dwellers who overwhelmingly depend on fishing (and farming), are heavily reliant on government and NGO-provided social safety allowances.. It is within this backdrop that that the Ashrayan (shelter/accommodation) project continued to expand, bolstered by the GoB's 2001 National Land Use Policy, which offers guidelines for improved land-use and zoning regulations (LANDac, n.d). In early 2018, the GoB formed a 10-member committee to assess if Bhasan Char was suitable for Rohingya relocation; the plan for relocation to the island was announced at the High Level Event on the Global Compact on Refugees in New York (Asia News Network, 2018).

*Enter Bhasan Char*
Bhasan Char ('floating island') previously known as *Char Piya* and *Thengar Char,* is an island in the Bay of Bengal, approximately 30 nautical miles from mainland Chittagong.[v] With an elevation of only 56 feet above sea-level, it is vulnerable to cyclones and submergence especially during high tide – a reality for coastal Bangladesh. Contrary to assumptions about Bhasan Char as 'an empty space,' or a deviation developed *only* for Rohingya management, the island has had a small local population living on its shores since its emergence.[vi] It was also the most recent addition to the GoB's Ashrayan Initiative, following Ashrayan 1, phase I (1997-2002); Ashrayan 1, phase II (2002–2010); Ashrayan 2: 2010–2022 to accomodate the country's landless populations, climate migrants, and more recently, the third-gender, Dalits, and Harijan communities.

In 2015, under its Ashrayan 3 Initiative, the GoB proposed relocating the Rohingya who had remained in Bangladesh following previous flows to Bhasan Char. Following the 2017 exodus, the GoB revisited the plan to include the newest arrivals in the relocation plan  In 2020, the Foreign Minister stated:

> The area where the Rohingya are concentrated is only 6800 acres. Intensive rainfall in this an area causes landslides – and there is always the possibility of Rohingya dying in such an event. *Then the international community will blame us*. Bhasan Char in contrast is a beautiful place. The Rohingya can be involved in farming and raising cattle… that is why we are trying our best to relocate them there (Maksud, 2020).

Furthermore, he attested that this decision did not preclude use of the Char as a tourist resort or as a place of residence of the country's chronically unhoused population (Kawser, 2020). In fact, some of the plans for Bhasan Char are similar to measures adopted by the GoB for making chars more livable for Bangladeshi nationals (Banerjee, 2023).[vii] The use of islands to provide housing for our poor, landless communities through the Ashrayan Initiativehas been an incredibly successful development project by government,' explained an official with the Ashrayan-2 initiative. 'The use of the Char is not an outlier, but is merely stage three of the Ashrayan Initiative – except that it now also includes the Rohingya (Interview, 2022).

With a budget of more than $350 million, the then GoB tasked the Bangladesh Navy with developing Bhasan Char. Ashrayan-3 comprises of semi-permanent shelters with the possibility of absorbing up to 400,000 people in the future (YouTube, 2019). Many low-lying areas were raised with an additional flood defence embankment to protect the island from environmental disasters (Ibid). In 2020, the government instructed all district administrations to send people from low-income groups in Bangladesh to Bhasan Char to make use of the island's facilities, as part of its 'return to home' initiative for those unable to maintain their livelihood in urban areas (Majumder, 2020). As of September 2024, the island has a total population of 35,629, mainly comprising of FDMNs from the 2017 exodus; although some Rohingya from the 1990s flow have also been relocated there 1990s e (UNHCR, 2024).[viii] In addition to the Rohingya there are approximately 2,000 people, inclusive of the host community, government employees, and NGO professionals currently residing in Bhasan Char.

Contextualizing Bhasan Char's within Bangladesh's political economy of traditional land use and its history of social safety schemes adds a different layer of nuance to understanding the island not as a offshoring project as in the case for instance of Nauru, but as an extension of the existing Kutapalong-Balukhali camps. However, several observations of refugee camps remain relevant. At one level, Bhasan Char conforms to the dynamics of inclusion and exclusion and of exceptional space that defines refugee camps. The constant surveillance of access and exit from the island through an endless bureaucratic maze closely monitored and surveilled by militarized personnel generates a Foucauldian notion of carcerality that challenges Turner's descriptor of camp porosity. At another level, Bhasan Char is not solely occupied by the Rohingya, which adds nuance to the citizen/outsider dichotomy that exists in a closed camp, given that Bangladeshi citizens exist in that space as economic, social, *and* surveillance actors. Furthermore, the small local economies and the space of interlocution between Bengalis, local and international aid workers and refugees while not an exceptional example, challenge the notion of absolute exclusivity. Such a space then, produces two distinct realities. From the perspective of the host state and even the previously skeptical international humanitarian community, Bhasan Char is a place of refugee autonomy devoid of enclosure with greater access to economic opportunities sans overcrowding. At another level, given that the *island itself is the camp*, this autonomy needs to be examined as an enclosed autocratic space, where refugee agency and everyday life is negotiated within carefully drawn parameters, and where governance occurs through hybrid sovereignty as a result of negotiations between the state and the international humanitarian industry. In this light, Bhasan Char remains a site for the arbitrary exercise of power and institutionalized imprisonment by the state and the humanitarian industry writ large.

### 7. A Triumph of (Local) Offshoring? Winners and Losers of the Bhasan Char Equation

In early 2019, recognizing that the move to Bhasan Char was imminent, the UNHCR Representative stated, 'we welcome the GoB proposal to de-congest the camps. We want to support the GoB in this initiative' (Prothom Alo, 2019). However, in the early days of implementation, the initiative came under severe international criticism, because of the environmental vulnerability of the island and concerns that relocation was initiated without full transparency (Human Rights Watch, 2021). International aid agencies also ignored an initial government invitation for a guided tour of the island. The U.K. Foreign, Commonwealth and Development Office released a statement noting, 'we have been clear with the UN and others that without independent, full and detailed assessments to make sure any refugees living there will be safe, this is not an option for any of our funding' (Ahasan, 2020). Such criticisms met with fierce rebuttal from the GoB, with authorities stressing relocation was an 'internal' affair. In a 2019 press conference, then PM Sheikh Hasina stated:

> Some aid organisations are opposing the move as [the] tourist district of Cox's Bazar is a nice place to for them to stay…some people are very interested to assist Rohingyas though they are not interested enough in repatriation of Rohingyas to their land' (New Age, 2019).

Following the first two rounds of relocation, which was criticized for not being fully voluntary, the Foreign Ministry further dismissed the accusations in a statement saying: 'It is disheartening that instead of appreciating the good intention of Bangladesh' (Majumder, 2020). An interview at the P-4 Humanitarian Relief Coordination office stressed,

> one cannot overlook the fact that working in Bhasan Char poses different set of challenges  and an adjustment for international aid workers. Yet, one questions why this island relocation is subject to so much controversy, particularly given that millions of our people have been living on chars. There is far more investment in Bhasan Char for the Rohingya than for Bangladeshis living in many of our islands..(interview, 2022).

Similar frustrations were also expressed with civil society and government actors pointed out that several of the same donor countries criticizing Bhasan Char were building walls and actively engaged in refugee pushback at their borders. Several interviewees also underscored how Bangladesh had continued to host – *not* deport - the Rohingya, and was establishing a livable space for the Rohingya with economic and educational opportunities.. In a 2020 Prothom Alo op-ed Maqsud argued... 'If the international community would work toward [the safe and sustainable return of the Rohingya] instead of stressing about Bhasan Char, the Rohingya community and Bangladesh would be extremely grateful.'

Initially, the GoB shouldered the full financial costs of developing Bhasan Char and relied on approximately 22 local NGOs to resettle the Rohingya on the island. The 2020 UNHCR MoU following intense negotiations was a turning point for international acceptance of the island relocation plan. The GoB also allowed international and national media access, and initiated a slew of diplomatic visits to the island including those by the ambassadors of the EU, South Korea, Japan, China, France, Germany, Denmark, Norway, the Canadian High Commissioner, the Swedish Envoy, the Danish Ambassador and the Chargé d'Affaires of the United States Embassy, the UNHCR Deputy High Commissioner, and the UNHCR Director for Asia and the Pacific. These efforts produced some positive results. For instance, while acknowledging he did not personally visit the island, the UNGA President Bozkir stated, 'I really applaud the work done there…I think this will be *another example to the world* on how to deal with refugee issues' (United News of Bangladesh, 2021). In a signal of support, the government of Japan became the first donor to commit to Bhasan Char's maintenance, contributing a $2 million grant to UNHCR-Bangladesh and the WFP. The UNHCR Representative in Bangladesh noted, 'This first ever funding by a donor to the UN operations on Bhasan Char signals solidarity with the Government and people of Bangladesh' (Business Standard, 2024).

Nevertheless, international financial contributions to Bhasan Char programming have been slow, with donors still expressing hesitation with Bhasan Char. In 2021, the Foreign Minister stated that the GoB would be requesting 10% of the funding for the Rohingya insisting

> it should not be the headache of the humanitarian agencies…whether Rohingyas are
> living in Kutupalong, Cox's Bazar, Barisal or in Bhasan Char. Their headache should be
> providing services to Rohingyas. They're obligated to give them services wherever they
> stay' (New Age, 2021).

An international humanitarian worker observed the 'dilemma' facing the international community in facing the demands for more assistance, noting: '…relocated Rohingyas are unlikely to get sufficient support despite growing demands on the island over time; yet external funding can lend legitimacy to the island project' (Interview, 2022). Even in 2022, the UN-led appeal for the Rohingya response, which included $100 million in funding for Bhasan Char did not receive significant support; the UK committed to funding the initiative for the first time, but the US – the largest donor to the Rohingya response - stated it 'does not currently support Bhasan Char' (Loy, 2022). This challenge to secure funding remains. In 2023, the government submitted two proposals to representatives from 17 countries and agencies, requesting assistance for more infrastructural assistance and programming, emphasizing that the Rohingyas will get a better life in Bhasan Char. Thus far, in 2024, the US has only committed 87 million of the $876 million Joint Response Plan in humanitarian aid to support Rohingya refugees and host communities in Cox's Bazar *and* Bhasan Char (USAID, 2024). According to UNHCR (2024), the significant shortfall between ongoing budgetary needs and operational expenditures underscore how the Rohingya continues to be to be primarily Bangladesh's responsibility.

While an in-depth discussion about the changing conditions on Bhasan Char is outside of the scope of the paper, it is important to consider Rohingyas' experiences with the relocation and its aftermath. The the GoB had insisted that relocation was in accordance with the MoU. Local researchers and NGO workers interviewed also insisted that the conditions on the island allow for increased mobility, better security, and more opportunities for the Rohingya. There is a growing body of scholarship and reports that highlight the current stability of the island, the opportunities it offers, and Rohingya satisfaction with the relocation (Gazi et. al, 2022; Islam, et. al 2021; Prothom Alo, 2022; ReliefWeb 2023). Critics have however argued that extensive PR campaigns have been instrumental in trying to create a 'positive' image for both international and domestic consumption. Reports have also noted the scarcity of verified independent information about the island, and how conversations with the refugees have been carefully curated within the framework of international diplomatic visits (Rahman, 2023; Daily Star, 2021; Devi, 2022). The reality that since 2021 several Rohingya have fled Bhasan Char in search of a future reveal ongoing challenges on the island and unresolved questions of a future for the Rohingya.

## 8. Conclusion

This research set out to explain the reasons as to why 100,000 Rohingya refugees in Bangladesh were slated to be moved to Bhasan Char – a remote island off the coast of Bangladesh. The study found that international indifference, failed attempts at repatriation, local and national frustrations with a protracted refugee situation, the perception of the Rohingya as a multidimensional threat – political, economic, environmental, demographic, and spatial – together with the historical and political economy of public land use contributed significantly to the GoB's decision for refugee relocation. Questions of performative security aside, the politics of land use offer important insights into why specific locations may be selected by Global South contexts to contain refugees. In particular, the research draws attention to both the socioeconomic history and politics of land distribution in Bangladesh and the Ashrayan Initiative – to contextualize why island relocation was seen as a feasible

response to concerns about camp congestion. While sufficient evidence was not found to argue conclusively that population relocation may serve Bangladesh's specific geostrategic interests vis-à-vis security in the Bay of Bengal, nevertheless, one of the main contributions of this research is that island relocation cannot be examined without considering the political economy of land use in the country where land concerns remain a site of intense contestation.

Second, this research draws attention to the need for more theorizing on how securitization dynamics emerge in Global South contexts. In Bangladesh, the transformation of the Rohingya as a 'hard' security issue can be explained in terms of strategic efforts to draw waning international attention to the protracted crisis, transforming yesterday's 'victims' to today's 'aggressors.' Yet, their securitization is not limited to questions of economics and politics – yet again, their construction as a threat is inextricably linked to questions environmental fragility and land politics in a context that remains at the forefront of the global climate crisis.

At a broader level, the research draws attention to how migration diplomacy across power differentials may be an unsuccessful venture for countries that are geostrategically unimportant and host refugees of 'less political value' to the west, and who do not engage in blackmailing, backscratching or mass expulsion strategies to leverage power in the international system.. In Bangladesh's case, drawing on the security and unsustainable argument, the calculation of using an island with significant investment was aimed at generating international accolade and continued assistance. While today Bhasan Char is touted as a 'model' of containment (with caveats) Bangladesh has not been able to instrumentalize it for concrete financial dividends. As the research has outlined, lingering questions about the island's feasibility, Bangladesh's limited leverage with a refugee population of 'low' political value, and competing geopolitical interests have all contributed to such a reality. Beyond the issue of miscalculation and financial 'loss,' however, the outcome of legitimization of refugee relocation in remote spaces also raises questions about how the search of the 'model' for refugee containment in places like Uganda and Bangladesh ultimately offers pathways for the Global North to continue to absolve themselves of equitable responsibility in protracted refugee crises.

**Acknowledgements**

I would like to thank all those without whom this project would not have been possible. The interest and extensive feedback I received for this research project through the Migration Politics Residency Program at the University of Glasgow have been nothing short of extraordinary. In particular, I would like to thank Gerasimos Tsourapas, Nick Macinski and Evelyn Ersinilli for their invaluable guidance, feedback and the wonderful conversations that emerged in discussing this research. In addition, I am deeply grateful to Darshan Vigeswaran, Benjamin Etzold, and Anas Ansar for their interest and insightful comments on my earlier drafts. I would like to thank Guadalupe Chavez for her kindness and company as a fellow resident during the residency program at Glasgow. My SIS research assistant Monzima Haque provided excellent research support at multiple stages of this project. I am grateful to my anonymous reviewers, as well as the editor, for their detailed and thoughtful comments on this paper and my interviewees in Bangladesh whose insight and expertise helped me understand the shifts and dynamics of Rohingya reception in Bangladesh.

**Funding information** This project was made possible through the American Institute of Bangladesh Studies (AIBS) Senior Fellowship Program.

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
