# Peer review of "The Curious Case of Bhasan Char: Island Relocation and the Politics of Refugee Containment in the ‘Global South’: The Case of Bangladesh"

_Migration Politics_

## Round 1 · Referee Report · Anonymous (Referee 1) · 2025-3-20

Strengths
Weaknesses
This sets up the second area for revision: the author needs to explain directly why this is a case of unsuccessful refugee rentierism. It is suggested several times throughout the text but it should be the main hook for the article because it shows why this manuscript is a contribution to the refugee rentierism literature.
The manuscript could be shortened, particularly in the history sections, and expanded in the theoretical sections above. I look forward to reading the revised manuscript as this is a serious and important contribution to our field.
Report
A few specific questions/ comments: - Don’t need as much explanation of single case methodology - Over use of ‘single’ quotes when it is not a direct quote of something p. 5 – “powerful myth” – please explain more. Myth for and by whom? This language might introduce more confusion if it is not the key concept of the paper. p. 5 – Not clear about the subheading “Where is the Global South?” -Section 4 (p. 7-10) is long and perhaps too much background. Consider shortening. -p. 12 - This sentence is not clear: “By erroneously assuming full availability of threat-related information and decisions within the public domain, it overlooks how in the Global South broadly, such information may not be publicly accessible; and how decision-making regarding security threats is characterized by less systematic and more ad hoc processes.” p. 13 – This sentence is not clear: “Post-British independence, land-based power politics was shaped by asymmetrical ‘patron-client’ or ‘headman-subordinated follower’ relations (Zaman, 1996).” How does this connect to your argument? p. 15 – “FDMNs” is confusing. Perhaps use a different phrase? p. 15 – “notion of absolute exclusivity” This phrase needs more explanation. It is the first time it is mentioned in the paper. p. 15 – “Bhasan Char remains a site for the arbitrary exercise of power and institutionalized imprisonment by the state and the humanitarian industry writ large.” This is a big claim. I agree with this analysis but it needs to be explained more and shown with evidence. p. 16 – “Similar frustrations were also expressed with civil society and government actors pointed out that several of the same donor countries criticizing Bhasan Char were building walls and actively engaged in refugee pushback at their borders.” This is an excellent point and should be expanded in the intro and conclusion. p. 17 – “the US has only committed 87 million of the $876 million Joint Response Plan in humanitarian aid to support Rohingya refugees and host communities in Cox’s Bazar and Bhasan Char”. This is not clear—Is the US committed to funding all of the JRP or is the 87 million the US contribution to a total amount of 876 for all donors?
Recommendation
Ask for minor revision

---

## Round 1 · Referee Report · Alice Nah (Referee 2) · 2025-5-2

Strengths
This is a paper with a lot of promise.
- The author writes well.
- The politics on the use of Bhasan Char for Rohingyas is a contentious issue worthy of deeper analysis.
- The protracted hosting of Rohingyas in Bangladesh is a strong case study for assessing refugee protection issues (and migration politics) in a Global Majority context.
Weaknesses
2. The author hasn’t provided the right theoretical framework and introduces a number of irrelevant debates and observations that muddy the contribution of the paper. I would suggest that the author choses one or two debates and follows them through in a deeper way, explaining their significance.
To this end, the author might want to consider comparing Bangladesh to other Global Majority situations that are similar – where the low-income states are unable to host refugee populations and requires international support, land is scarce, and it has to appease and look after citizens. Drawing on literature and concepts from very different contexts (often European) does not help the author to pinpoint the precise contribution of the paper. More relevant examples might be sourced from Africa and Asia.
The author raises a couple of interesting points that could be explored further. Section 6 on the political economy of land use and “char living” is unique and interesting. The practices describe here sound like internal colonialism – the classifying of land as ‘new land’, taking control of it, ignoring existing native populations, and inviting and relocating different groups to the island. The Rohingyas are possibly (disposable?) ‘early settlers’ to demonstrate that char living possible for citizens. There is something going on here in Bangladesh’s intentions and actions that are not fully explored and explained.
3. The author underplays the deep tensions between Bangladesh and the international community in the relocation of Rohingyas to Bashan Char. More in-depth analysis is needed of the concerns raised by other governments and civil society groups (about environmental degradation, the safety and security of refugees), as well as Bangladesh’s insistence and persistence in relocating Rohingyas in the face of such opposition.
Report
Requested changes
Not applicable.
Recommendation
Reject

---

## Round 2 · Author Response

Thank you for your thoughtful comments on my manuscript. I appreciated the opportunity to revise the manuscript substantively based on the central concern - that the theoretical contribution of the paper needed to be more developed and clearly stated. The paper no longer explains the reasons for the choice of Bhasanchar. Instead, the revised submission has a clear research question and is grounded in the scholarship of migration diplomacy, in particular the literature on refugee rentierism. It now analyzes the extent to which the refugee rentierism model is limited in explaining Bangladesh's refugee relocation plan, and offers a constructivist expansion that takes into account the country's attempt at what I identify as 'norm modeling.' The article has also also been substantively edited to remove the discursive and description sections that are tangential to the discussion - in particular the sections on refugee camps, global refugee protection trends, and securitization of refugees in the Global South. The section on the background of the Rohingya has been streamlined and the discussion of the earlier attempts at their repatriation has been notably reduced. The paper also offers the term internal offshoring and ties it more clearly to the realities of land use, in particular the use of islands in Bangladesh, taking into account the history and tensions around the use of land in the country. The last two sections of the paper revisit the core argument and highlight what is hoped to be an important contribution to the existing literature on refugee rentierism. Last, but not the least, the remaining typos and infelicities of language have been removed.

---

## Round 2 · List of Changes

The paper has gone undergone significant revision. Below are the changes that have been made: 1. A new abstract 2. A significantly revised introduction with a clear research question. 3. The paper is no longer about reasons for the island relocation, but examination of the decision based on existing scholarship on migration diplomacy. 4. Grounding of the research question in migration diplomacy scholarship, particularly the literature on refugee rentierism. 5. Introduction of the concepts of internal offshoring and normative modeling as constructivist modification to the concept of refugee rentierism. 6. Definition of internal offshoring and tying the concept more clearly to the realities of land use in Bangladesh. 7. The discursive and description discussions of several issues particularly that of refugee camps and global refugee protection trends have been removed. 8. The discussion on Bangladesh's engagement with regional and international agreements, and the 'indifference' discussion has been significantly condensed. 9. The table on agreements has been moved as an annex. 10. The discussion on the Rohingya has been notably shortened. 11. The discussion on security and securitization has been shortened significantly since it is now more tangential to the core question. 12. The internal offshoring discussion focusing on Bangladesh's land use and challenges has been expanded. 13. The 'winners and losers' section has been revised. 14. The conclusion was significantly revised to emphasize the core question and summary of the theoretical contributions of the article was provided.

---

## Editorial Decision

unknown